# Glycogen Synthase Kinase-3 Inhibition by CHIR99021 Promotes Alveolar Epithelial Cell Proliferation and Lung Regeneration in the Lipopolysaccharide-Induced Acute Lung Injury Mouse Model

**DOI:** 10.3390/ijms25021279

**Published:** 2024-01-20

**Authors:** Raquel Fernandes, Catarina Barbosa-Matos, Caroline Borges-Pereira, Ana Luísa Rodrigues Toste de Carvalho, Sandra Costa

**Affiliations:** 1Life and Health Sciences Research Institute (ICVS), School of Medicine, University of Minho, Campus Gualtar, 4710-057 Braga, Portugal; pg36886@alunos.uminho.pt (R.F.); id8801@alunos.uminho.pt (C.B.-M.); id7788@alunos.uminho.pt (C.B.-P.); 2ICVS/3B’s—PT Government Associate Laboratory, 4806-909 Braga, Portugal; 3Department of Internal Medicine, São João Universitary Hospital Center, 4200-319 Porto, Portugal; ana.toste@chsj.min-saude.pt; 4Faculty of Medicine, University of Porto, 4200-319 Porto, Portugal

**Keywords:** acute lung injury, glycogen synthase kinase-3, alveolar epithelial cells, cell proliferation, lung regeneration

## Abstract

Acute respiratory distress syndrome (ARDS) is a life-threatening lung injury that currently lacks effective clinical treatments. Evidence highlights the potential role of glycogen synthase kinase-3 (GSK-3) inhibition in mitigating severe inflammation. The inhibition of GSK-3α/β by CHIR99021 promoted fetal lung progenitor proliferation and maturation of alveolar epithelial cells (AECs). The precise impact of CHIR99021 in lung repair and regeneration during acute lung injury (ALI) remains unexplored. This study intends to elucidate the influence of CHIR99021 on AEC behaviour during the peak of the inflammatory phase of ALI and, after its attenuation, during the repair and regeneration stage. Furthermore, a long-term evaluation was conducted post CHIR99021 treatment at a late phase of the disease. Our results disclosed the role of GSK-3α/β inhibition in promoting AECI and AECII proliferation. Later administration of CHIR99021 during ALI progression contributed to the transdifferentiation of AECII into AECI and an AECI/AECII increase, suggesting its contribution to the renewal of the alveolar epithelial population and lung regeneration. This effect was confirmed to be maintained histologically in the long term. These findings underscore the potential of targeted therapies that modulate GSK-3α/β inhibition, offering innovative approaches for managing acute lung diseases, mostly in later stages where no treatment is available.

## 1. Introduction

Acute respiratory distress syndrome (ARDS) is a severe form of acute lung injury (ALI) resulting from a severe inflammatory reaction of the lungs to pulmonary damage, with mortality rates ranging from 30% to 50% depending on the severity of the disease [1,2]. Primarily triggered by direct lung insults such as pneumonia (30–50%), or indirect systemic insults such as sepsis (20–30%) and shock, ARDS inflicts damage upon the alveolar epithelium or the alveolar-capillary barrier, respectively [3,4]. The initial phase of ARDS is characterised by an exacerbated production of inflammatory cells and pro-inflammatory cytokines and chemokines, inflammatory cell infiltration, diffuse alveolar epithelium damage, cell death, and disruption of the alveolar-capillary barrier, culminating in lung edema and compromised gas exchange [5,6]. The removal of dead cells, re-epithelialisation of the alveolar population, and restoration of the endothelial barrier are key processes during the recovery phase that culminate in lung regeneration and repair [7,8]. While some patients can recover naturally or after supportive care and pharmacological interventions in the initial phases targeting the initial insult, a significant gap persists in clinical treatments directed towards the pathophysiological mechanisms of ARDS. Patients who are unable to undergo efficient lung repair and regeneration tend to develop interstitial pulmonary fibrosis, which prolongs ventilator dependence, leading to restrictive lung disease later in life, and carries high morbidity and mortality rates [9].

Under normal circumstances the adult lung has a remarkable capacity to regenerate after injury [10], with regionally distinct epithelial adult stem cells mediating homeostasis and regeneration [11,12,13,14,15,16,17]. Nevertheless, the human lung is subject to degenerative disease and in the face of grave insults, the homeostatic and reparative mechanisms may no longer be adequate to maintain its structure, integrity, and function.

Alveolar epithelial cell regeneration is a crucial process in lung repair after injury [8], but the exact mechanisms underlying this process in ARDS are poorly understood. A better understanding of how the lung regenerates after ARDS is essential for the identification of novel therapeutical targets and molecules to enhance naturally occurring repair mechanisms.

Recent studies using a directed-differentiation protocol of human pluripotent stem cells (hPSCs) toward lung epithelium cells have shown that glycogen synthase kinase-3 (GSK-3) inhibition by CHIR99021 promotes lung progenitor proliferation, while GSK-3 inhibition withdrawal induces differentiation of airway epithelium [18,19,20] and alveolar epithelial cell type II (AECII) maturation [19,20]. Another study has shown that timed GSK-3 inhibition withdrawal induces differentiation of AECII cells from lung progenitors, after which re-exposure to CHIR99021 drives AECII cell expansion [21]. 

These studies highlighted GSK-3 as a pivotal player in critical regenerative mechanisms such as proliferation and differentiation [22,23,24,25,26,27,28]. GSK-3 is a serine/threonine protein kinase with two isoforms, GSK-3α and the GSK-3β encoded by two different genes [29]. It is involved in multiple signaling pathways, and its activity is finely modulated by phosphorylation, protein complex association, and subcellular localisation [30]. Beyond its roles in glycogen synthesis [31], hematopoietic stem cell self-renewal [23], and cellular processes, including cell proliferation, differentiation, and death hematopoietic stem cell self-renewal [23], recent insights highlight its association with inflammatory responses via nuclear factor kappa light chain enhancer of activated B cells (NF-kB) [32,33,34]. In the lung, GSK-3 has been shown to play a crucial role in the maturation of lung progenitors and proximal-distal specification through the modulation of NOTCH and WNT signaling in a 3D collagen model of human airway lineage specification [20]. Furthermore, studies have demonstrated GSK-3’s involvement in modulating AMP-activated protein kinase (AMPK) inactivation in lung injury, which is involved in anti-inflammatory functions [35]. Additionally, it activates NF-kB, thereby reducing the downstream expression of pro-inflammatory genes [36]. GSK-3 regulates extracellular matrix remodeling and fibroblast-to-myofibroblast transition upon TGF-β stimulation, which is crucial for lung repair [37,38]. Furthermore, a study of ARDS induction in mice demonstrated increased expression and interaction of ELAVL-1/HuR and GSK-3β, suggesting the interaction between GSK-3β and ELAV-1 during ARDS [39].

Some studies have shown that GSK-3 inhibition in ALI models promotes a decrease in the production of pro-inflammatory cytokines [36], an increase in anti-inflammatory cytokines [34], attenuates the accumulation of pulmonary edema [40], increases mouse survival in a lipopolysaccharide (LPS)-induced endotoxemia model [41], and improves lung repair and function [42]. Of particular note, the GSK-3 inhibitor CHIR99021 has demonstrated specificity, potency, and a unique ability to inhibit both GSK-3α and GSK-3β isoforms, setting it apart from other inhibitors [20,43]. Moreover, CHIR99021 is essential for maintaining the alveolar lineage in alveolar progenitors [44,45], making it a valuable tool for investigating the effects of GSK-3 in lung regeneration. Nevertheless, the effects of GSK-3α/GSK-3β inhibition in lung homeostasis and injury, namely acute lung injury (ALI), remain unclear.

In the quest to understand the impact of GSK-3 inhibition by CHIR99021 in AEC regeneration in ALI, this study used the LPS-induced ALI model, targeting the acute inflammatory phase characterised by a pro-inflammatory response driven mainly by the innate immune system and the late stages characterised by repair and regeneration processes [9]. Therapeutic interventions in the late stages are almost noneffective in targeting with faster disease progression and higher mortality and morbidity [9]. Considering this, we conducted a pilot study to determine whether late administration of CHIR99021 could be used as a therapeutic alternative to target the late phase of LPS-induced ALI.

Our findings revealed that GSK-3α/β inhibition with CHIR99021, particularly during the late phase, sparks a recovery in alveolar epithelial populations, promoting their proliferation and transdifferentiation. This suggests the potential contribution of CHIR99021 to improve lung regeneration. Additionally, we conducted a pilot study to dissect the effect of CHIR99021 on the inflammatory response. In contrast to the results obtained in studies using other GSK-3 inhibitors, our data seem to indicate that CHIR99021 does not impact inflammation when administered both at early and late stages of LPS-induced ALI. This suggests that CHIR99021 may not directly modulate the mechanisms involved in inflammation. However, definitive conclusions on this topic needs further investigation. Given the lack of effective treatments during the repair phases of acute lung diseases, these findings open up new possibilities for more effective therapies for ARDS and related conditions.

## 2. Results

### 2.1. GSK-3 Inhibition by CHIR99021 Attenuates Lung Injury in LPS-Induced ALI

Lung injury is a pathological feature of human ARDS and LPS-induced ALI models [5]. First, we evaluated the impact of CHIR99021 on tissue damage during the early phase marked between 12 h and 24 h after LPS-induced ALI when immune cells have a predominant role [46,47]. For that purpose, we analysed lung injury at 24 h post-administration of LPS and CHIR99021 treatment at 30 min post-LPS through hematoxylin and eosin (H&E) staining. Moreover, CHIR99021 was administered at 18 h post-LPS injection to assess its potential effects in the late phases of the disease. Lung injury was assessed by H&E staining at 72 h after LPS-induced ALI, a late phase of the disease marked by an attenuation of the inflammatory cells [46,47], and after 60 days, a repair phase. An attenuation in immune infiltration, alveolar congestion, hyaline membranes, and alveolar wall thickness was observed after CHIR99021 treatment at the time points mentioned (Figure 1a). Moreover, quantification of the lung injury score showed that GSK-3 inhibition by CHIR99021 attenuated histological lung injury at the early and late phases of LPS-induced ALI (Figure 1b).

Alveolar–capillary barrier damage is another indicator of lung injury in ARDS and the LPS-induced ALI mouse model [5]. The heightened permeability of the barrier allows for the accumulation of high molecular weight proteins and fluid in the interstitium, contributing to impaired gas exchange [5]. Accordingly, we investigated whether GSK-3 inhibition by CHIR99021 at 24 h post-LPS and CHIR99021 treatment at 30 min post-LPS injection and 72 h post-LPS and CHIR99021 treatment at 18 h post-LPS injection affected alveolar–capillary barrier permeability (Appendix A). This assessment was conducted through the Evans blue dye test and wet-to-dry lung weight ratio at the earlier inflammatory response time (Appendix A) and by measuring the total protein content present in bronchoalveolar lavage fluid (BALF) at the later inflammatory response time (Appendix A). LPS administration increased Evans blue and total protein concentrations in BALF compared to animals that received PBS, highlighting that LPS induces damage to the alveolar–capillary barrier (Appendix A). Conversely, no significant differences were observed after CHIR99021 treatment at both time points analysed. These data imply that CHIR99021 does not impact the damage to the alveolar–capillary barrier.

Apoptosis is a critical process of programmed cell death evident in the pathophysiology of human ARDS and the early phase of ALI [48,49,50,51]. Furthermore, studies have demonstrated that LPS induces lung epithelial and endothelial cell damage, ultimately leading to the apoptosis of these cells [52,53]. The presence of apoptotic cells in diffuse alveolar damage has been identified, indicating that the increase in apoptosis reflects the degree of lung damage in ALI [52,54]. Considering these findings, we assessed the protein levels of cleaved caspase-3, a component of the apoptotic cascade, at 12 h and 24 h post-LPS administration, along with CHIR99021 treatment at 30 min. Additionally, DNA fragmentation was evaluated at 24 h using the terminal deoxynucleotidyl transferase dUTP nick end labelling (TUNEL) assay. As expected, the expression of apoptotic readouts is higher in the animals treated with LPS than in the control animals treated with PBS (Figure 2). Interestingly, at 12 h and 24 h, respectively, administration of CHIR99021 decreased 59% and 77% of the expression of cleaved caspase-3 compared to the LPS group untreated with CHIR99021 (Figure 1a,b). Moreover, at 24 h, the expression of cleaved caspase-3 in animals treated with CHIR99021 showed no difference to the level observed in the control group, suggesting a return of apoptosis to baseline levels, while LPS-induced injury in untreated animals was still ongoing. The expression of TUNEL-positive cells decreased by 70% in the LPS group treated with CHIR99021 compared to the LPS group untreated (Figure 2c,d). These results indicate that GSK-3 inhibition by CHIR99021 mitigates cell apoptosis during the early phase of LPS-induced ALI, reflecting the alleviation of lung injury. 

Taken together, these findings revealed that GSK-3 inhibition by CHIR99021 during the early and late phase of LPS-induced ALI attenuates lung injury.

### 2.2. GSK-3 Inhibition by CHIR99021 Does Not Affect the Recruitment of Inflammatory Cells and the Production of Pro- and Anti-Inflammatory Cytokines in LPS-Induced ALI

Some studies in ALI models using GSK-3 inhibitors other than CHIR99021 have shown that GSK-3 inhibition attenuates lung damage and increases lung repair by attenuating the immune response and increasing anti-inflammatory mediators [36,41,42,50,55]. Regrettably, various anti-inflammatory drugs tested in clinical trials have not successfully improved lung repair or the prognosis of ARDS patients [56,57]. This presents a critical gap in effective pharmacological therapies for ARDS, leading to poor quality-adjusted survival rates among survivors, often associated with issues like muscle dysfunction, reduced attention, memory, concentration, and executive function [58]. These challenges have increased the interest in understanding the mechanisms underlying lung regeneration and repair, a pivotal process observed in ARDS patients who recover. 

Recent studies in human pluripotent stem cell-derived lung progenitors demonstrated the impact of CHIR99021 on lung progenitors, affecting their distal/proximal maturation [18,20,21]. This prompted the hypothesis that GSK-3 inhibition via CHIR99021 may directly influence lung progenitors, fostering lung regeneration and repair. To substantiate this hypothesis, we first decided to evaluate whether the observed reduction in lung injury following CHIR99021 treatment was due to direct effects on the inflammatory response. For that, we analysed the recruitment of immune populations by flow cytometry and the protein levels of pro-inflammatory (IL-1β, IL-6, and TNF-α) and anti-inflammatory cytokines (IL-10) by ELISA in BALF at 12 h post-LPS and CHIR99021 treatment at 30 min post-LPS injection, and 30 h and 72 h post-LPS and CHIR99021 treatment at 18 h post-LPS injection. 

Our analysis indicated that administrating CHIR99021 at 30 min did not influence the recruitment of leukocytes, neutrophils, Ly-6C^+^ monocytes, interstitial macrophages, T cells, CD4^+^ T cells, and CD8^+^ T cells (Figure 3a) at 12 h after LPS-induced ALI. Similarly, no significant differences were observed in the expression of the pro-inflammatory cytokines IL-1β, IL-6, and TNF-α (Figure 3b), and in the anti-inflammatory cytokine IL-10 (Figure 3b) following CHIR99021 treatment at this time point.

Administration of CHIR99021 at 18 h post-LPS injection, after the subsiding of the immune response, also revealed no alterations in the recruitment of leukocytes, neutrophils, eosinophils, Ly-6C^+^ monocytes, alveolar macrophages, interstitial macrophages, T cells, and CD4^+^ T cells at 30 h and 72 h post-LPS-induced ALI (Figure 3a). Comparably, there were no significant differences in the expression of the pro-inflammatory and anti-inflammatory cytokines analysed after CHIR99021 treatment at these time points (Figure 3b).

Consequently, our data seem to reveal that GSK-3 inhibition by CHIR99021 does not significantly affect the inflammatory response in the early and late phases of LPS-induced ALI. Therefore, the observed attenuation in lung injury following CHIR99021 treatment does not appear to be directly associated with reduced inflammation.

### 2.3. GSK-3 Inhibition by CHIR99021 Promotes the Proliferation of Cells Expressing AECI and AECII Markers in LPS-Induced ALI

A study by Carvalho et al. [20] displayed that GSK-3 inhibition by CHIR99021 influences lung progenitors and alveolar maturation of human pluripotent stem cell-derived lung progenitors. Understanding the pivotal role of cell proliferation and differentiation in the re-epithelisation of the alveolar population, essential for lung regeneration and repair, prompted our exploration of the impact of CHIR99021 on cell proliferation, particularly on AECs. We measured cyclin D1 protein levels, a marker of proliferation, by western blot at 12 h and 24 h post-LPS injection and CHIR99021 treatment at 30 min post-LPS injection, and 30 h and 72 h post-LPS injection and CHIR99021 treatment at 18 h post-LPS injection. Additionally, at 24 h and 72 h, we conducted immunofluorescence studies using proliferating cell nuclear antigen (PCNA), podoplanin (PDPN), and prosurfactant protein C (proSP-C) to identify the populations with increased proliferative levels after early or late administration of CHIR99021: AECI (PDPN positive), AECII (proSP-C positive), or cells expressing both AECI and AECII markers, indicating a potential transition stage between the alveolar lineages or alternatively, a bipotential alveolar progenitor [12,59].

We observed that CHIR99021 treatment at 30 min and 18 h post-LPS injection resulted in increased cyclin D1 levels at the time points studied (Figure 4a,b) compared to LPS-treated animals untreated with CHIR99021. Moreover, at 24 h, we noted an increase in the expression of double-positive PCNA/proSP-C cells (Figure 4c,d) and triple-positive PCNA/PDPN/proSP-C cells (Figure 4c,d) in animals treated with CHIR99021, suggesting that early CHIR99021 administration promotes the proliferation of AECII cells and cells expressing both AECI and AECII markers, respectively. Notably, at 72 h, there was a massive increase in the expression of PCNA (Figure 4c,d) and triple-positive PCNA/PDPN/proSP-C cells (Figure 4c,d) after CHIR99021 treatment at 18 h post-LPS injection.

These findings indicate that GSK-3 inhibition by CHIR99021 in the late-stage injury promotes the proliferation of a subpopulation of AECs expressing both AECI and AECII markers. Moreover, the results support the impact of CHIR99021 in attenuating lung injury, as observed in histological data. 

### 2.4. GSK-3 Inhibition by CHIR99021 Promotes Alveolar Epithelial Population Renewal in LPS-Induced ALI

The previous sections have showcased that early or late administration of CHIR99021 mitigates lung injury and cell apoptosis without significantly altering the inflammatory response. Furthermore, CHIR99021 stimulates AECII proliferation during the early stages of LPS-induced ALI. Nevertheless, it prompts the proliferation of cells expressing both AECI and AECII markers upon early and late administration. These observations imply that CHIR99021 alleviates the pathological features of LPS-induced ALI by stimulating the proliferation of alveolar epithelial cells. However, it remains crucial to discern whether these features are linked to restoring the damaged alveolar epithelium, pivotal for lung regeneration, repair, and maintenance of lung function.

To address this query, we conducted western blot analyses of various markers at 24 h post-LPS injection and CHIR99021 treatment at 30 min post-LPS injection, and 30 h and 72 h post-LPS instillation with CHIR99021 administered at 18 h post-LPS injection. These markers included homeobox only protein x (HOPX), responsible for AECII to AECI transdifferentiation [17,60], along with proSP-C, PDPN, and AQP5 (a marker of mature AECI). Cells expressing both proSP-C and PDPN markers could potentially lead to mature AECII or AECI [8,59,61]. Furthermore, we previously showed, by immunofluorescence analysis, that at 24 h and 72 h, there is tissue expression of both proSP-C and PDPN double-positive cells (Figure 4c,d).

Administration of CHIR99021 at 30 min post-LPS injection promoted an increase in proSP-C at 24 h post-LPS (Figure 5a,b). Moreover, treatment with CHIR99021 at 18 h promoted an increase in the protein levels of HOPX and proSP-C at 30 h and proSP-C, PDPN, and AQP5 at 72 h (Figure 5a,b). Late CHIR99021 administration enhanced AECII and AECI populations at 72 h post-LPS administration (Figure 5c,d). Additionally, we observed an increase in the expression of markers associated with cells expressing both AECI and AECII markers at 72 h (Figure 5c,d).

These outcomes indicate that GSK-3 inhibition by CHIR99021 late in the time course of injury promotes the transdifferentiation of AECII into AECI, potentially linked to an increased proliferative capacity of AECII and the augmentation in AECI number. Furthermore, the increase in AECII cell markers and the presence of a subpopulation of AEC double-positive for AECII/AECI markers suggests that GSK-3 inhibition by CHIR99021 could contribute to the renewal of the alveolar epithelial population and, consequently, lung regeneration in ALI later stages.

## 3. Discussion

In ARDS, direct or indirect insults often compromise the alveolar epithelium and the alveolar–capillary barrier, significantly impacting lung function [4]. Alveolar epithelium restoration is a crucial process observed in patients who successfully recover from this condition [10]. Notably, the absence of effective treatments, particularly during the late phases and for severe forms of the disease characterised by a failure in repair mechanisms, poses a substantial challenge.

In the last decades, studies have been conducted to elucidate the populations of lung epithelial cells that modulate lung injury and repair, as well as the underlying mechanisms of such responses. To date, AECII stand out as they have been extensively investigated and are proposed as the progenitors of the adult lung alveolus [62]. AECII play a pivotal role in surfactant synthesis, secretion, and recycling. Research has demonstrated their capacity to proliferate and differentiate into AECI in injury contexts, a process crucial for maintaining lung integrity [45,63]. Interestingly, the modulation of alveolar epithelial progenitor proliferation and maturation through GSK-3 inhibition by CHIR99021 in human pluripotent stem cell-derived lung progenitors has been evidenced [20]. However, the impact of GSK-3 inhibition by CHIR99021 in ALI models remains unexplored. Consequently, our study delved into the role of GSK-3 inhibition by CHIR99021 in promoting epithelial cell proliferation and regeneration.

The LPS-induced ALI model was employed to mimic the main features of human ARDS [64]. To explore the impact of GSK-3 inhibition during the predominant phase of inflammatory cell presence corresponding to the early stage of LPS-induced ALI, we administered CHIR99021 30 min after intratracheal LPS injection. Moreover, to assess GSK-3 inhibition when inflammatory cells were subdued, we administered CHIR99021 18 h after LPS injection and pointed out its potential contribution to effective targeted therapies effective for late stages of the disease, which nowadays are lacking. Using H&E staining, a lung injury score calculation, considering the alveolar congestion, immune infiltration, macrophage and neutrophil counts, hyaline membrane formation, and alveolar wall thickness [65,66], revealed that both early and late CHIR99021 administration effectively attenuated lung injury. Existing studies indicated a decrease in neutrophils 24 h post-LPS and the completion of the inflammatory phase by 72 h [46,47]. However, our histological analysis illustrated a substantial accumulation of neutrophils and macrophages at 72 h post-LPS. The flow cytometry analysis conducted corroborated these observations. Notably, the attenuation in lung damage after CHIR99021 administration in the late phase of the model persisted for 60 days post-LPS-induced ALI, during which the accumulation of neutrophils and macrophages seemed to decrease. These findings underscore the potential of GSK-3α/β inhibition to modulate the later phases of LPS-induced ALI.

GSK-3 holds a crucial role in innate immunity by promoting the production of pro-inflammatory cytokines following TLR stimulus [34]. Studies using GSK-3 inhibitors other than CHIR99021 have demonstrated the attenuation of the inflammatory response and increased anti-inflammatory cytokine production [35,36,50,55]. However, our research found that both early and late administration of CHIR99021 did not significantly affect immune cell recruitment or cytokine production. Minimal differences were observed in the populations of eosinophils and alveolar macrophages between the LPS-treated and untreated CHIR99021 groups. However, these populations had relatively low percentages compared to other immune populations in the lung post LPS-induced ALI. Similarly, minimal differences were noted in the CD8^+^ T cell population in low percentages. Thus, our findings imply that CHIR99021 does not directly impact the immune response, and that the observed attenuation of lung injury induced by GSK-3 inhibition is not primarily promoted by immune cells.

The role of GSK-3 in cell death, proliferation, and differentiation has been established [22,23,30,67,68,69]. Moreover, studies have outlined apoptotic mechanisms post-LPS treatment [49,70,71], where apoptosis inhibitors mitigated several ALI features, such as pulmonary edema, neutrophil accumulation, and epithelial cell death [52,53]. Our study further demonstrated that CHIR99021 significantly attenuated apoptosis during the early phase of LPS-induced ALI, highlighting a potential mechanism by which CHIR99021, and thus GSK-3α/β inhibition, contributes to reduced lung injury, which is then followed by enhanced lung regeneration.

Studies have demonstrated the effectiveness of GSK-3β inhibition in mitigating lung injury, cell death, and inflammation, along with the underlying mechanisms. Studies using SB216763 showed a reduction in lung injury, BAL neutrophils, and macrophages, pro-inflammatory cytokines, apoptosis, total protein levels in BALF, lung interstitial exudation, and alveolar rupture [54,55,72,73]. While some studies did not delve into the mechanisms underlying these effects, others suggested that GSK-3 inhibition activates WNT/β-catenin signaling, thus retarding inflammatory cell infiltration and cytokine production [54,55,73]. Similar outcomes were corroborated using genipin, with authors proposing that inflammation and oxidative stress attenuation are modulated by the NF-kB/ nuclear factor erythroid 2-related factor 2 (Nrf2) mechanism [74] and TDZD-8 via inhibiting the activation of NF-kB signaling [36,75]. Park, D.W. et al. [35] demonstrated GSK-3β’s contribution to inhibited AMPK, enhancing LPS-induced inflammatory responses. They illustrated that GSK-3β inhibition with 6-bromoindirubin-3′-oxime (BIO), SB216763, or siRNA knockdown reduced the expression of pro-inflammatory cytokines, preventing the inhibition of AMPK [35]. Other studies explored the effects of other GSK-3β inhibitors in lung injury. The results demonstrated an attenuation of lung injury and the production of pro-inflammatory mediators [50,76] by modulation of WNT signaling, NF-kB, guanine nucleotide exchange factor-H1 (GEF-H1)/ROCK signaling that regulates β-catenin [40], and the TGF-β-GSK-3β megalin axis [42]. However, in these studies, GSK-3 inhibitors were administered prior to the induction of lung damage. This approach does not accurately replicate the development of the human disease, so the results were not discussed.

Contrary to other GSK-3 inhibitors that selectively target only one of the GSK-3 isoforms, CHIR99021 distinguishes itself by inhibiting both isoforms. This unique characteristic modulates and activates different signaling pathways, indirectly influencing the expression of pro- and anti-inflammatory cells and molecules. Consequently, the observed lack of impact of CHIR99021 on inflammation, when compared to findings from studies using other GSK-3 inhibitors, can be attributed to its distinct mechanism of action.

Given the critical importance of cell proliferation and differentiation of AECII to AECI transdifferentiation after ALI, we then explored whether the lung injury attenuation post-CHIR99021 treatment could be linked to these crucial mechanisms involved in lung regeneration. Several studies have demonstrated the effect of CHIR99021 in stimulating proliferation, observed in lung cancer cell lines [77], lung progenitor cells in pulmonary fibrosis [78], radiation models [79], and neural progenitor cells [80]. Our findings indicated that early and late GSK-3 inhibition by CHIR99021 significantly increased cell proliferation. Additionally, we revealed that early CHIR99021 treatment stimulated AECII proliferation and led to increased proliferation of cells expressing both AECI and AECII markers at early and late phase CHIR99021 administration. The exact mechanism by which GSK-3α/β inhibition promotes AECII proliferation is not known. It is possible, however, that modulation of WNT plays a role, as canonical WNT signaling inhibits GSK-3 Ser/Thr kinase activity, thereby stabilising β-catenin [81], and it has been reported that WNT signalling promotes proliferation of a subset of AECII involved in alveologenesis and alveolar repair [13].

We then hypothesised that the continued proliferation of cells expressing AECI and AECII markers at 72 h post-LPS administration indicated ongoing regenerative processes. We investigated whether the increased proliferation rates of alveolar progenitor cells post-CHIR99021 treatment were correlated to increased AECII-to-AECI transdifferentiation and the regeneration of the alveolar epithelial population. We found that late CHIR99021 treatment increased the expression of PDPN, an immature AECI marker, along with proSP-C concentration at the late stage of LPS-induced ALI. Moreover, late CHIR99021 administration increased HOPX expression, a marker of AECII-to-AECI transdifferentiation [17,60]. This aligned with the increase in PDPN and AQP5, a mature AECI marker, during the late post-LPS injection period. Jansing et al. [61] similarly demonstrated alveolar epithelial transdifferentiation after LPS-induced ALI, characterised by the downregulation of AECII-related markers and the upregulation of AECI-related markers. Our findings imply that the observed increase in the AECII marker proSP-C could indicate ongoing, yet incomplete, regeneration. 

Bronchoalveolar stem cells (BASCs) are vital in lung regeneration post-injury. These cells can self-renew and differentiate into AECI, AECII, club, and ciliated cells [82,83]. BASCs also co-express AECII markers such as proSP-C [84], with staining revealing differences from AECII by the absence of lamellar bodies. Our data indicated that cells labelled with pro-SPC displayed the morphological appearance of inclusion bodies in the cytoplasmic region, which is characteristic of AECII [85]. Moreover, several signaling pathways promoting the transition of AECII to AECI have been elucidated. Recent characterisation of AECII-derived intermediate cells, emerging during reparative stages after bacteria-induced damage, identified distinct AECII subgroups, supporting the hypothesis of AECII differentiation towards AECI [86]. This intermediate state was characterised by the expression of keratin (Krt) 8, *Cldn4*, *Ctgf*, and *Sfn* [86]. Studies have suggested the involvement of immune cells and inflammatory paracrine factors such as TNF-α and IL-1β in ACEII repair and transition into AECI [87,88]. Our preliminary results on the inflammatory niche demonstrated that these cells are not affected by CHIR99021, suggesting that this inhibitor does not modulate inflammation-mediated AEC transition. Other studies described that growth factors, including fibroblast growth factor (FGF) and epidermal growth factor (EGF), promote AECII proliferation [12,89]. Furthermore, it has been demonstrated that bone morphogenetic protein (BMP) 4 signaling [90], downregulation of NOTCH activity [91], downregulation of β-catenin and WNT signaling [92], and upregulation of the Hippo pathway [93], are needed for AECII transition into AECI. On the other hand, studies have shown that AECI can exhibit increased plasticity during regenerative processes and is capable of proliferating and differentiating into AECII [17]. In this case, after lung injury, Hopx^+^ insulin-like growth factor-binding protein 2 (Igfbp2)^−^ AECI maintains cellular plasticity and may participate in repair and regeneration by transdifferentiating into AECII [94]. The mechanisms underlying these processes are unknown. Some studies demonstrated that AECI and AECII arise from a bipotent alveolar progenitor cell during lung development, which expresses both AECI- and AECII-related markers [36,59]. Additionally, data have identified a subset of AECII with increased co-expression of SFTPC and *Ager*, AECII and AECI markers, after lung injury [90]. These findings align with Evans’ original observation that AECII proliferate and give rise to transitional cells with the co-expression of AECII and AECI phenotypes after lung injury [95]. Jansing et al. [61] demonstrated that AECII-to-AECI transdifferentiation during repair after lung injury involves initial cell spreading, and studies demonstrated that traced AECI give rise to AECII with co-expression of AECI and AECII markers during murine lung development [17] and in human lungs [96] through the inhibition of TGF-β signaling. Interestingly, our findings showed that a subpopulation of AECs, with the expression of the same markers of a foetal bipotent alveolar progenitor population, AECI (PDPN) and AECII (proSP-C) markers, present a vast proliferation and a highly increased number in the ALI later stage studied. Thus, our data suggest that an alveolar epithelial progenitor population described in foetal lung development could be a candidate to contribute significantly to lung regeneration in ALI. Nevertheless, further investigation is necessary to understand the exact contribution of different lung epithelial cell populations with stem/progenitor cell capacities to post-ALI regeneration. Some mechanisms and signaling pathways involved in lung regeneration after lung injury and GSK-3 inhibition were discussed. Thus, further studies involving lineage tracing mouse models subjected to LPS-induced ALI and then treated with CHIR99021 might bring some enlightenment into the latter.

## 4. Materials and Methods 

### 4.1. Mice

Male C57BL/6J:C3Heb/FeJ mice, maintained inbreed from their colony, were used. All experiments were conducted in accordance with the European Union Directive 2010/63/EU and were approved by the local ethics committee (SECVS 032/2017). The animals were housed under specific pathogen-free conditions, adhered to a 12:12-h light–dark cycle, were kept at 22 °C, 55% humidity, and fed with a standard diet and water ad libitum.

### 4.2. Experimental Design

Animals were distributed homogenously into three experimental groups, taking into consideration their weight. The experimental design is depicted in Figure 6.

### 4.3. LPS-Induced ALI Mouse Model

Male mice 7 to 9 weeks old were intratracheally injected with an optimized single dose of 5.0 mg/kg of lipopolysaccharide based on literature evidence [97] (LPS; Escherichia coli O111:B4, Calbiochem, Sigma-Aldrich (St. Louis, MO, USA), 437627) dissolved in 1x phosphate–buffered saline (PBS). 30 min or 18 h after administration of LPS or PBS, animals were intraperitoneally injected with 2.0 mg/kg of CHIR99021 based on literature evidence [79] (Merck (Darmstadt, Germany), Sigma-Aldrich, SML1046) in 7.5% dimethyl sulfoxide (DMSO) (Sigma-Aldrich, D2650) to evaluate its effects during the peak of the inflammatory cell recruitment (early injury phase) or its attenuation (late injury phase), respectively. Analgesic was administered every 12 h until the end of the experiments, and 20% weight loss and respiratory failure during or after surgery were established as human endpoints. 12 h and 24 h after administration of LPS and CHIR99021 treatment at 30 min or 30 h and 72 h after administration of LPS and CHIR99021 treatment at 18 h, animals were humanely sacrificed by cervical dislocation or anaesthesia overdose.

### 4.4. Histological Examination

Animals were sacrificed by anaesthesia overdose and transcardially perfused with 1x PBS for the analysis of lung injury. The lungs were dissected, and one-third of each lobe for histological and immunofluorescence analysis was fixed in 4% paraformaldehyde (PFA) for 24 h at room temperature (RT), embedded in paraffin, and cut into 3–5 μm-thick-sections. The sections used for histological studies were stained with H&E, and two researchers independently evaluated the slides in a double-blind manner. About 20% of the total lung tissue per section of each animal was photographed in the Olympus BX61 light microscope (Olympus BX61 Upright Microscope, Tokyo, Japan) coupled with Olympus DP70 digital camera using a magnification of 400×. The lung injury score was quantified considering alveolar wall thickness, congestion of the alveoli, immune cell infiltration, number of macrophages and neutrophils, and formation of hyaline membranes. The semiquantitative score, ranging from zero to twenty-three, was independently assessed by two different individuals in a blinded manner (Table 1). Two independent experiments were conducted and subjected to statistical analysis. Since both experiments yielded statistical differences, the data from one of them are presented herein.

### 4.5. Immunofluorescence

After deparaffinisation for 1 h at 60 °C, sections were hydrated, and then antigen retrieval on a microwave (600 W) using 10 mM citrate buffer at pH 6.0 was performed. The slides were incubated with 0.1% Triton X-100 (Honeywell Fluka (Morris Plains, NJ, USA), 31434), 0.1% sodium citrate permeabilization solution during 30 min at RT and blocked with 5% bovine serum albumin (BSA) (Sigma-Aldrich, A3294) for 1 h at RT. Next, the sections were incubated with PCNA (Sigma, Millipore (Burlington, MA, USA), AB9260, 1/200), PDPN (R&D Systems (Minneapolis, MN, USA), AF3244, 1/50), and proSP-C (Sigma, AB9260, 1/400) overnight (ON) at RT, washed with 1x PBS-0.05% Tween (Tw), incubated with secondary biotin-conjugated (Sigma, Millipore, B6649, 1/100), Alexa Fluor^®^ 647 (Thermo Fisher Scientific (Waltham, MA, USA), A21245, 1/150), and Alexa Fluor^®^ 488 (Thermo Fisher Scientific, A11055, 1/150) antibodies for 2 h at RT. Then, the slides were incubated with streptavidin Alexa Fluor^®^ 568 conjugate antibody (Thermo Fisher Scientific, S11226, 1/100) for 1 h at RT, washed with 1x PBS-0.05% Tw, and mounted with PermaFluor™ Aqueous Mounting Medium (Thermo Fisher Scientific, TA-030-FM). The images were acquired on the Olympus LPS Confocal FV1000 (Olympus U-TBI90, Japan) at 400x magnification and processed by the FV10-ASW, Version 04.02.01.20 (David Berneda and Marc Meumann) software.

The TUNEL assay using an in situ cell death detection Kit (Roche (Basel, Switzerland), 11684817910) was performed following the manufacturer’s protocol. Slides were mounted with PermaFluor™ Aqueous Mounting Medium, and images were acquired on the Olympus BX61 light microscope (Olympus, Japan) coupled to the Olympus DP70 digital camera and processed by the Olympus cellSens Dimension 1.18, Olympus Corporation© software.

Two independent experiments were conducted and subjected to statistical analysis. Since both experiments yielded statistical differences, the data from one of them are presented herein.

### 4.6. Bronchoalveolar Lavage Fluid Collection

For BALF collection, a catheter was inserted into the exposed trachea, the lungs were instilled with around 1.5 mL of 1x PBS, and then the liquid was aspirated and repeated three times. The BALF was centrifuged at 1500 rpm for 5 min at 4 °C. The supernatant was stored at −80 °C and then used for permeability analysis and quantification of IL-1β, IL-6, TNF-α, and IL-10 cytokines. The pellet was used for flow cytometry analysis (explained in Section 4.7). Two independent experiments were conducted and subjected to statistical analysis. Since both experiments yielded statistical differences, the data from one of them are presented herein.

### 4.7. Flow Cytometry

Animals were over-anesthetised by intraperitoneal injection, the BALF was collected as detailed above, and then the animals were perfused. The lungs were incubated with 1 mL of DNase I (Roche, 1010415900) and collagenase II (Gibco (Waltham, MA, USA), 17101-015) solution (1 mg/mL DNase I and 5 mg/mL Collagenase II in Dulbecco’s Modified Eagle Medium–DMEM, Gibco, 3121969-035), minced into pieces between 2–3 mm^3^, and enzymatically and mechanically digested. The pellet from BALF was added to the tissue suspension of each animal and incubated with ammonium-chloride-potassium (ACK) lysis buffer (Gibco, A1049201) for 4 min at RT to lyse red blood cells and then incubated with the double volume of 1x PBS. 

Next, 1 × 10^6^ cells of each single cell suspension were loaded into a cytometer tube and incubated with TruStain FcXTM (anti-mouse CD16/32) antibody (Biolegend (Tokyo, Japan), 101320) for 20 min at 4 °C. After this period, the samples were incubated for 30 min at 4 °C in the dark with the fluorochrome-conjugated primary antibodies: CD45-BV510 (Biolegend, 103138, 1/100), F4/80-PerCP-Cy 5-5 (Biolegend, 123128, 1/100), CD11c-BV605 (Biolegend, 117333, 1/100), CD11b-PE-Cy7 (Biolegend, 101216, 1/100), Ly-6G-APC-Cy7 (BD Biosciences, 560600, 1/300), Ly-6C-BV711 (Biolegend, 128037, 1/100), SiglecF-PE (BD Biosciences, 552126, 1/200), CD64-FITC (Biolegend, 139316, 1/300), CD8-BV785 (Biolegend, 100749, 1/100), (Biolegend, 100545, 1/100), CD3-APC (Biolegend, 100236, 1/100). Then, samples were incubated with viable dye (VD) eFluor^®^ 450 (EBioscience (San Diego, CA, USA), 65-0863-14) for 30 min at 4 °C in the dark. At the end, the samples were incubated with 4% PFA for 20 min at 4 °C. The next day, cells were analysed using the LSR II flow cytometer system (BD Bioscience (Franklin Lakes, NJ, USA)). 300,000–500,000 events were obtained per sample. Data were analysed in FlowJo 10.7.1 software (Becton Dickinson & Company (BD), Ashland, OR, USA). The autofluorescence background from the antibodies was corrected using unstained control. Single stained controls for each antibody were performed using AbC Total Antibody Compensation Beads Kit (Life Technologies, Carlsbad, CA, USA, A10497), submitted to the same staining protocol as the samples. The gating strategy used to identify the immune cell populations is described in Appendix A.

### 4.8. ELISA

Sandwich Enzyme-Linked Immunosorbent assay (ELISA) kits were performed to detect the concentrations for IL-1β (BioLegend’s ELISA MAXTM Deluxe Set, cat n° 432604), IL-6 (BioLegend’s ELISA MAXTM Deluxe Set, cat n° 431304), TNF-α (BioLegend’s ELISA MAXTM Deluxe Set, cat n° 430904), and IL-10 (Thermo Fisher, cat n° 88-7105-88). The protocols were performed according to the manufacturer’s instructions. An automated microplate reader (Varioskan^®^ Flash-Thermo Fisher Scientific, Vantaa, Finland) at 450 nm was utilised to read the absorbance. The absorbance at 570 nm was subtracted from the absorbance at 450 nm to eliminate contaminations. The concentrations were calculated based on the standard curve run on each assay plate. Two independent experiments were conducted and subjected to statistical analysis. Since both experiments yielded statistical differences, the data from one of them are presented herein.

### 4.9. Western Blotting

Protein extraction was performed with lysis Buffer (LB) solution (50 mM Tris-base, 5 mM ethylenediaminetetraacetic acid (EDTA), 150 mM sodium chloride, 1.0% Triton X-100, and 1/7 protease inhibitor cocktail, diluted in ultrapure water). Total protein concentration was measured spectrophotometrically at 595 nm and calculated against a BSA standard curve in the microplate reader (Varioskan^®^ Flash-Thermo Fisher Scientific, Vantaa, Finland). Samples were prepared in the proportion of 1:1 with a loading buffer (950 μL of Laemmli Sample Buffer 2x (Biorad (Hercules, CA, USA), 161-0737) and 50 μL 2-mercaptoethanol (Merck, Millipore, 63689)), and denatured for 5 min at 98 °C. Then, samples were loaded in a 15% sodium dodecyl sulphate-polyacrylamide gel (SDS-PAGE) and transferred to Immobilon^®^-FL polyvinylidene fluoride (PVDF) membranes (Merck Millipore, IPFL 00005) or AmershamTM Protran^®^ nitrocellulose membranes (Sigma-Aldrich, GE10600008) using the Trans-Blot^®^ TurboTM Transfer System (BioRad Laboratories, München, Germany). Membranes were blocked in 5% BSA for 1 h at RT.

Following this, membranes were incubated ON at 4 °C with primary antibodies against cyclin D1 (Abcam (Cambridge, UK), ab16663, 1/200), cleaved caspase-3 (Cell Signaling Technology, #9661, 1/250), HOPX (Sigma, Millipore, HPA030180, 1/1000), proSP-C (Chemicon (Tokyo, Japan), AB3786, 1/1000), PDPN (R&D Systems, AF3244, 1/750), AQP5 (Abcam, ab78486, 1/2000), and β-actin (Abcam, ab8224, 1/10000), diluted in 5% Milk or 5% BSA according to the manufacturer’s instructions. After washing with 1x PBS-0.05% Tw, membranes were incubated with horse radish peroxidase (HRP)-coupled secondary antibodies mouse IgG (Santa Cruz Biotechnology (Dallas, TX, USA), 56102, 1/3000), goat anti-rabbit IgG (Cell Signaling Technology (Danvers, MA, USA), 7074-P2, 1/2000), donkey anti-goat IgG (Santa Cruz Biotechnology, 2020, 1/2500), and diluted in 5% Milk for 1 h at RT. Another washing step was performed, and antibodies were detected in Sapphire Biomolecular Imager (Azure Biosystems, Dublin, CA, USA) using the Western Bright™ Sirius HRP Substrate Kit (Advansta (San Jose, CA, USA), K-12043-D10). The bands were quantified in Image Lab™ software (Biorad, version 6.1.0) and quantifications were normalised to β-actin levels. Two independent experiments were conducted and subjected to statistical analysis. Since both experiments yielded statistical differences, the data from one of them are presented herein.

### 4.10. Statistical Analysis

Data were analysed in GraphPad Prism, version 8.4.0 (GraphPad Software, San Diego, CA, USA). Normality was measured by the Shapiro–Wilk statistical test and was assumed when *p* > 0.05. All data were evaluated using an unpaired *t*-test. When normality was achieved, a parametric *t*-test was used, and when normality was not achieved, the nonparametric Mann–Whitney test was applied. Outliers were identified and excluded, taking into consideration the ROUT method (Q = 10%). The results were presented as mean ± Standard Error of the Mean (SEM). The level of statistical significance was considered at *p* < 0.05.

## 5. Conclusions

Evidence suggests that CHIR99021, a GSK-3 inhibitor, acts as a modulator of alveolar epithelial lineages. Despite this, the impact of CHIR99021 on lung regeneration, a critical process in lung repair after ALI, remains unknown. For the first time, this study elucidates the influence of CHIR99021 on AEC during both the inflammatory and regeneration stages of ALI. We assessed the effects of early and late administration of CHIR99021 on lung injury through histology and cell death analysis, AEC proliferation, and repopulation of AEC populations. Additionally, a pilot study was conducted to investigate the effect of CHIR99021 on the immune response.

Our study outlines that the administration of CHIR99021 effectively mitigates lung injury and cell death, promotes alveolar cell proliferation, and stimulates the renewal of AECI and AECII, probably due to the enhancement of AEC proliferation and increase of an alveolar epithelial progenitor-like subpopulation. This effect is particularly pronounced when CHIR99021 is administered at the late phase of lung injury, when inflammatory cell activity is subdued.

This study marks the first comprehensive exploration into the role of GSK-3 inhibition by CHIR99021 in AEC proliferation following ALI, especially in the late phase of the disease when available therapies are lacking. Notably, later administration of CHIR99021 not only enhances AEC proliferation but also appears to alter the dynamics of AEC regeneration by boosting the proliferation of an AEC subpopulation expressing AECI and AECII markers, thereby promoting transdifferentiation and overall regeneration. The lack of effect on the immune response implies that CHIR99021, unlike other GSK-3 inhibitors, does not directly modulate inflammation-related mechanisms.

Our findings pave the way for novel therapeutic developments in lung injury, showing the use of GSK-3α/β inhibitors as a promising therapeutic strategy, notably in late-stage ARDS, when no effective treatment is available. Furthermore, in light of the AEC damage induced by SARS-CoV-2, the viral pathogen responsible for COVID-19 [86], the shared mechanisms observed in COVID-19-associated ARDS such as pronounced inflammation, lung injury, severe pulmonary edema, and epithelial repair [98,99], along with the existing scientific evidence highlighting the involvement of GSK-3 in the disease’s development, this study has the potential to open up new possibilities for utilising GSK-3 inhibition with CHIR99021 in COVID-19 patients. 

## Figures and Tables

**Figure 1 ijms-25-01279-f001:**
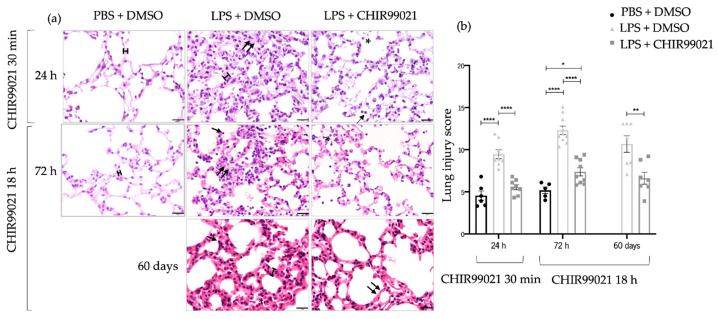
Early and late effect of GSK-3 inhibition by CHIR99021 on lung injury. (**a**) Lung sections of hematoxylin and eosin staining from animals at 24 h after intratracheal injection of LPS and CHIR99021 treatment at 30 min post-LPS injection (upper panel); 72 h after intratracheal injection of LPS and CHIR99021 treatment at 18 h post-LPS injection (intermediate panel), and 60 days after intratracheal injection of LPS and CHIR99021 treatment at 18 h post-LPS injection (bottom panel). (**b**) Quantification of lung injury score considering alveoli congestion, immune cell infiltration, the relative number of macrophages (single arrow), and neutrophils (asterisk), formation of hyaline membranes (double arrows), and the alveolar wall thickness (scale). 400× magnification. Scale bars: 20 μm. (n = 5–11 per group. N = 2 independent experiments. Data were presented from one of them, both with statistical differences). Data are present as mean ± SEM. * *p*-value < 0.05, ** *p*-value < 0.01, **** *p*-value < 0.0001, parametric *t*-test.

**Figure 2 ijms-25-01279-f002:**
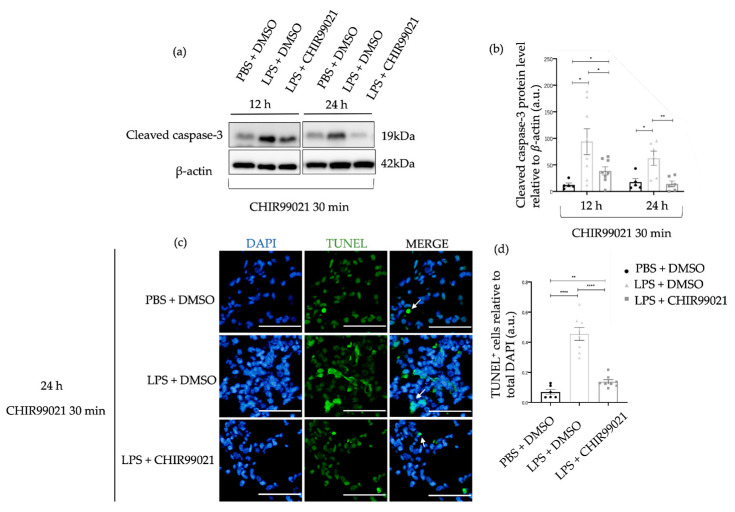
Effect of GSK-3 inhibition by CHIR99021 on lung cell apoptosis. (**a**) Representative images of western blot using lung cell suspensions at 12 h and 24 h after LPS-induced ALI and CHIR99021 treatment at 30 min post-LPS injection blotted with anti-cleaved caspase-3 and anti-β-actin (loading control). (**b**) Quantification of cleaved caspase-3 protein levels normalised to β-actin at 12 h and 24 h. (**c**) Representative images of TUNEL staining at 24 h for the different study groups. TUNEL-positive cells are shown in green and highlighted with white arrows. The mean total number of cells observed for the PBS + DMSO, LPS + DMSO, and LPS + CHIR99021 group were 803 cells, 845 cells, and 857 cells, respectively. 400× magnification. Scale bars: 50 μm. (**d**) Quantification of TUNEL positive cells relative to total DAPI cells. (n = 6–8 per group. N = 2 independent experiments. Data were presented from one of them, both with statistical differences). Data are present as mean ± SEM. * *p*-value < 0.05, ** *p*-value < 0.01, **** *p*-value < 0.0001, parametric and *t*-test. a.u.: Arbitrary units, DAPI: 4′,6-diamidino-2-phenylindole, TUNEL: Terminal deoxynucleotidyl transferase dUTP nick end labelling.

**Figure 3 ijms-25-01279-f003:**
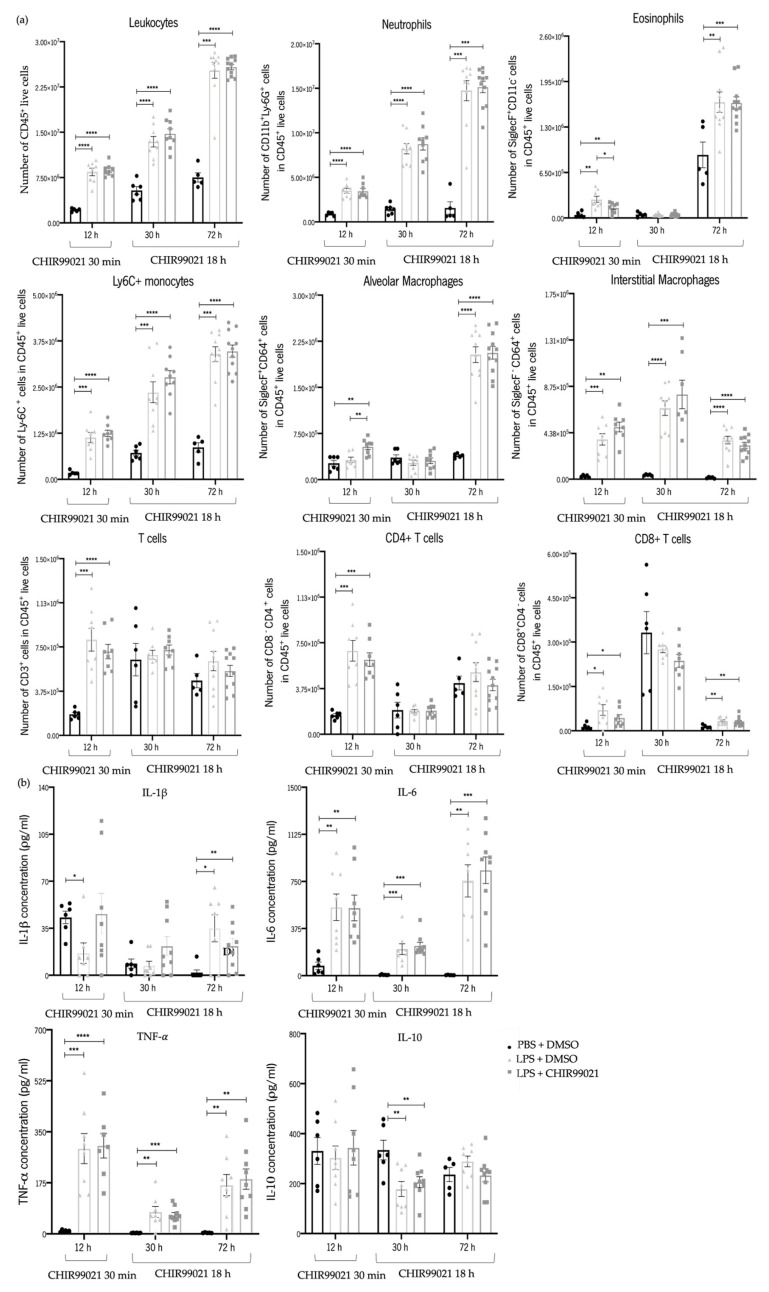
Early and late effect of GSK-3 inhibition by CHIR99021 on lung inflammation. Using flow cytometry, immune cell populations in mouse lungs were assessed at 12 h after intratracheal injection of LPS and CHIR99021 treatment at 30 min post-LPS injection and 30 h and 72 h after intratracheal injection of LPS and CHIR99021 treatment at 18 h post-LPS injection (**a**,**b**). (**a**) Analysed immune cell populations included: Leukocytes (CD45^+^), Neutrophils (CD11b^+^Ly-6G^+^), Eosinophils (SiglecF^+^CD11c^−^), Ly-6C^+^ monocytes (Ly-6C^+^), Alveolar macrophages (SiglecF^+^CD64^+^), Interstitial macrophages (SiglecF^−^CD64^+^), T cells (CD3^+^), CD4^+^ T cells (CD4^+^CD8^−^), and CD8^+^ T cells (CD4^−^CD8^+^), all cell populations analysed at 12 h after LPS injection and CHIR99021 treatment at 30 min, at 30 h after LPS injection and CHIR99021 treatment at 18 h, and at 72 h after LPS injection and CHIR99021 treatment at 18 h. (**b**) Quantification of pro-inflammatory cytokines including IL-1β, IL-6, TNF-α, and the anti-inflammatory cytokine IL-10 expression presented in BALF supernatant at the indicated time points (n = 5–11 per group. N = 2 independent experiments. Data were presented from one of them, both with statistical differences). Data are present as mean ± SEM. * *p*-value < 0.05, ** *p*-value < 0.01, *** *p*-value < 0.001, **** *p*-value < 0.0001, parametric and nonparametric *t*-test. IL: Interleukin, TNF: Tumour necrosis factor.

**Figure 4 ijms-25-01279-f004:**
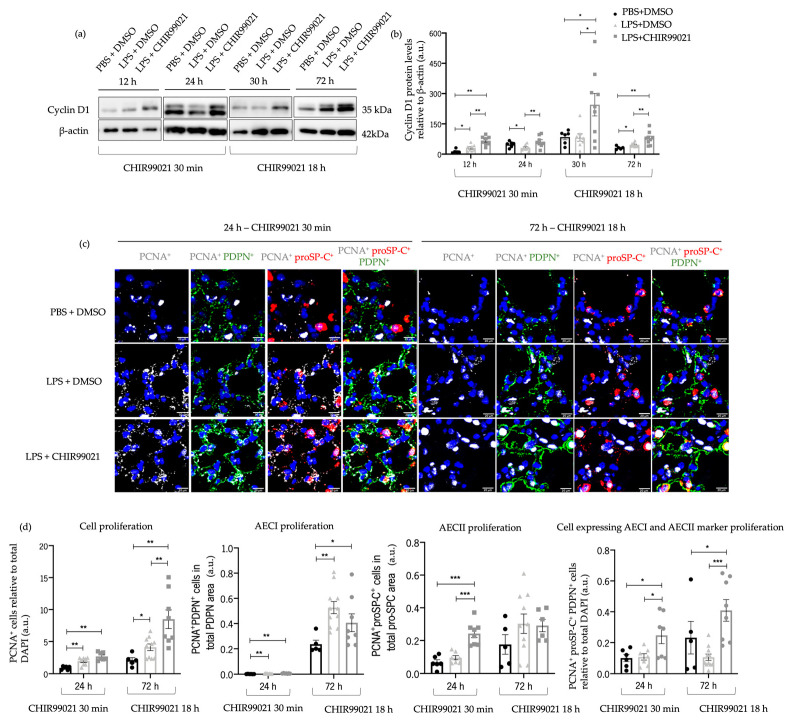
Effect of GSK-3 inhibition by CHIR99021 on lung proliferation. (**a**) Representative images of western blot using lung cell suspensions at 12 h and 24 h after LPS-induced ALI and CHIR99021 treatment at 30 min post-LPS injection and 30 h and 72 h after LPS-induced ALI and CHIR99021 treatment at 18 h post-LPS injection blotted with anti-Cyclin D1 and anti-β-actin (loading control). (**b**) Quantification of Cyclin D1 protein levels normalised to β-actin at 12 h, 24 h, 30 h, and 72 h. (**c**) Representative immunofluorescence images of PCNA, PDPN, and proSP-C expression at 24 h and 72 h. 400× magnification. Scale bars: 20 μm. (**d**) Relative quantification of total PCNA positive cells per total DAPI positive cells, total double-positive PCNA/PDPN cells per total positive PDPN area, total double-positive PCNA/proSP-C cells per total proSP-C area, and total triple-positive PCNA/proSP-C/PDPN cells per total DAPI positive cells, at 24 h and 72 h. (n = 5–11 per group. N = 2 independent experiments. Data were presented from one of them, both with statistical differences). Data are present as mean ± SEM. * *p*-value < 0.05, ** *p*-value < 0.01, *** *p*-value < 0.001, parametric and nonparametric *t*-test. a.u.: Arbitrary units, PCNA: Proliferating cell nuclear antigen, PDPN: Podoplanin, proSP-C: Prosurfactant protein C.

**Figure 5 ijms-25-01279-f005:**
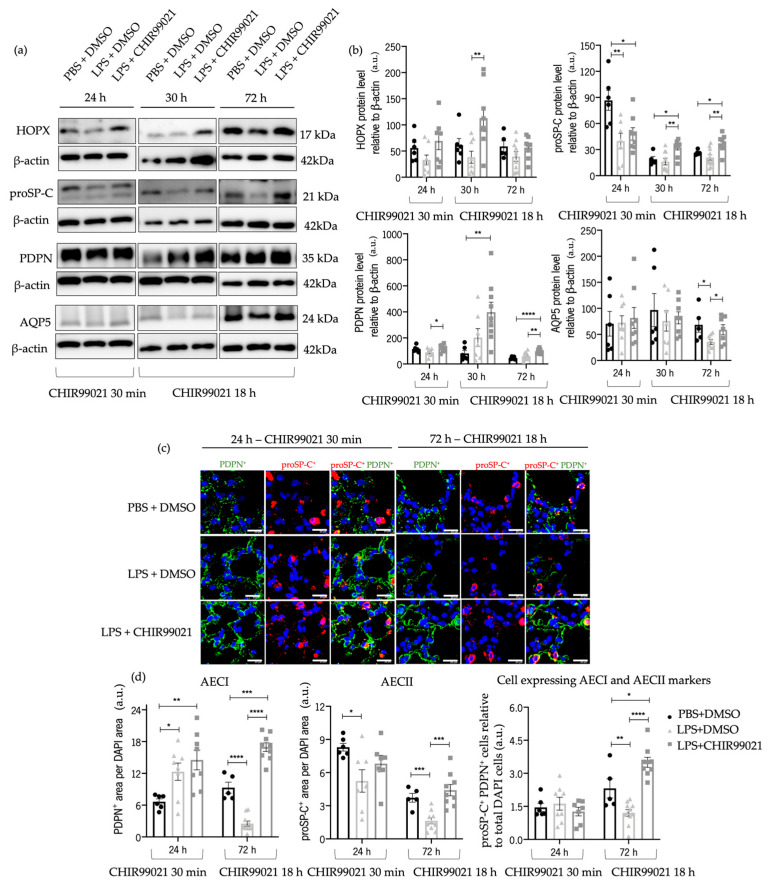
Effect of GSK-3 inhibition by CHIR99021 in AECI and AECII regeneration. (**a**) Representative images of western blot using lung cell suspensions at 24 h after LPS-induced ALI after CHIR99021 treatment at 30 min post-LPS injection, and at 30 h and 72 h after LPS-induced ALI after CHIR99021 treatment at 18 h post-LPS injection, blotted with anti-HOPX, anti-proSP-C, anti-PDPN, anti-AQP5, and anti-β-actin (loading control). (**b**) Quantification of HOPX, proSP-C, PDPN, and AQP5 protein levels normalised to β-actin at 24 h, 30 h, and 72 h. (**c**) Representative immunofluorescence images to PDPN and proSP-C at 24 h and 72 h. 400× magnification. Scale bars: 20 μm. (**d**) Relative quantification of PDPN positive area per DAPI area, proSP-C positive area per DAPI area, and double positive proSP-C/PDPN cells in total DAPI cells, at 24 h and 72 h. (n = 5–11 per group. N = 2 independent experiments. Data were presented from one of them, both with statistical differences). Data are present as mean ± SEM. * *p*-value < 0.05, ** *p*-value < 0.01, *** *p*-value < 0.001, **** *p*-value < 0.0001, parametric and nonparametric *t*-test. AEC: Alveolar epithelial cells, AQP5: Aquaporin 5, a.u.: Arbitrary units, HOPX: Homeobox only protein x, PDPN: Podoplanin, proSP-C: Prosurfactant protein C.

**Figure 6 ijms-25-01279-f006:**
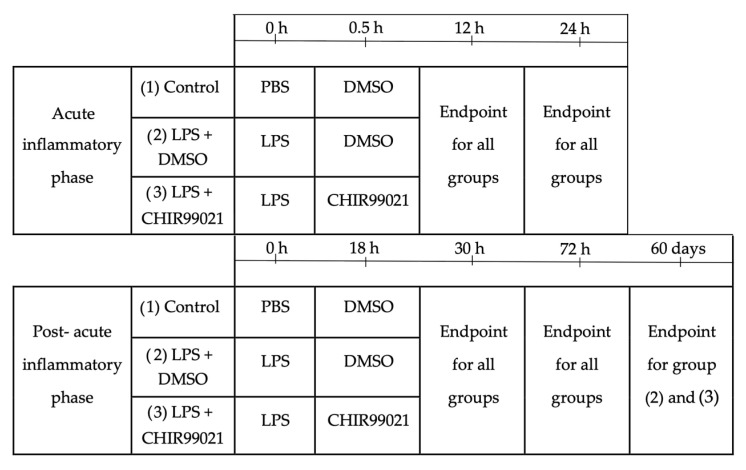
Experimental design. Male mice, aged 7 and 9 weeks, weighing between 20 and 28 g, were divided into three groups (PBS + DMSO, LPS + DMSO, LPS + CHIR99021), ensuring a consistent weight range across the groups. To assess the impact of CHIR99021 during the phase dominated by inflammatory cell activity, CHIR99021 or DMSO was intraperitoneally administered 30 min after intratracheal injection of LPS or PBS. Animals were humanely sacrificed 12 h or 24 h after the commencement of the experiment. To investigate the effects of CHIR99021 during the post-acute inflammatory phase, characterised by diminished inflammatory cell activity, CHIR99021 or DMSO was intraperitoneally administered 18 h after intratracheal injection of LPS or PBS. Animals were humanely sacrificed 30 h, 72 h, or 60 days after the start of the experiment.

**Table 1 ijms-25-01279-t001:** Grading criteria to define the lung injury score. Each parameter has a grading criterion according to the severity of lung injury. The total lung injury score was calculated in 20% of the total lung tissue per section of each animal using all the parameters.

Parameter	Score
Alveoli congestion	0: Minimal damage; 1: Mild damage 2: Moderate damage; 3: Severe damage; 4: Maximal damage
Immune infiltration	0: Minimal damage; 1: Mild damage 2: Moderate damage; 3: Severe damage; 4: Maximal damage
Number of macrophages	0: <5; 1: 5–10; 2: 10–20; 3: 20–50; 4: >50
Number of neutrophils	0: <5; 1: 5–10; 2: 10–20; 3: 20–50; 4: >50
Hyaline membranes	0: Minimal damage; 1: Mild damage 2: Moderate damage; 3: Severe damage; 4: Maximal damage
Alveolar wall thickness	0: equal to control; 1: 2x; 2: 2x–4x; 3: >4x

## Data Availability

The authors will freely release all data supporting the published paper upon direct request to the corresponding author.

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
