# Peer review of "Glycogen Synthase Kinase-3 Inhibition by CHIR99021 Promotes Alveolar Epithelial Cell Proliferation and Lung Regeneration in the Lipopolysaccharide-Induced Acute Lung Injury Mouse Model"

_ijms, 2024, doi:10.3390/ijms25021279_

Round 1

Reviewer 1 Report

Comments and Suggestions for Authors

This is a great review showing that the GSK-3 inhibitor CHIR99021 has a major impact on lung repair and regeneration upon LPS induced acute lung injury. The GSK-3 inhibitor promoted AECI and AECII proliferation and the AECII cell differentiation into AECI cells.  

Perhaps the author may state whether basal cells were altered and whether the respiratory epithelial integrity such as protein leak was affected by the inhbitor and ev consider a graphical abstract.

Reviewer 2 Report

Comments and Suggestions for Authors

While this study by Fernandes et al. further demonstrates the ability of the GSK-3 inhibitor CHIR99021 to mitigate structural lung damage and enhance epithelial progenitor cell proliferation in a mouse LPS model, major concerns exist regarding the depth of mechanistic investigation and functional assessments of injury and repair.

Few minor comments:

Introduction section

  1. The introduction provides good background on ARDS, but more information is needed on the specific mechanisms of action of GSK-3 in lung injury and repair to set up the rationale for studying CHIR99021.
  2. Additional references could be included when introducing GSK-3 and its involvement in regenerative mechanisms to strengthen that background.
  3. When introducing CHIR99021, the authors could briefly explain why this specific inhibitor was chosen over other GSK-3 inhibitors to study the impacts on lung progenitors.

Results section 4. In Figure 1, please include scale bars on the histology images.

  1. For Figure 2, please state in the legend the total number of cells quantified for the apoptosis analyses.
  2. Great demonstration overall of the anti-apoptotic effects of CHIR99021 during early stages. This is an important finding highlighted in the text as well.
  3. In Figure 3, clarify in the legend which timepoints the flow cytometry data in 3A corresponds to.
  4. The lack of impact of CHIR99021 on immune cell recruitment is an interesting and unexpected finding that contrasts some previous work. This point comes across clearly in the text.
  5. For Figures 4 and 5, please add units to the Y-axis labels (e.g. protein expression relative to β-actin).
  6. The increases in AEC proliferation markers in Figure 4 are clear and reinforce the impacts of CHIR99021 observed on histology.

Discussion section 11. The discussion could better incorporate how the findings fit with previous work using other GSK-3 inhibitors in lung injury models.

  1. When mentioning the lack of impact of CHIR99021 on inflammation, discuss why results may differ from other GSK-3 inhibitor studies.
  2. More speculation could be provided on the specific mechanisms behind CHIR99021's effects - e.g. specific signaling pathways modulated.

Materials and Methods 14. Please provide more specific details on the CHIR99021 and LPS doses chosen - e.g. cite prior studies.

  1. Were experiments replicated? If so, please indicate replicate number and how data across replicates was handled.
  2. For lung injury score criteria in Table 1, indicate whether score parameters were assessed across entire lung sections or just within imaged areas.
  3. Please include gating strategy for flow cytometry experiments, either as main or supplementary figure.
  4. For westerns, please provide catalog numbers for all primary antibodies used.

Figures 19. Increase font size in Figure legends to improve readability.

  1. Add scale bars to Figure 1 histology images.
  2. For Figures 4 and 5, add units to Y-axis labels.
  3. Standardize font styling and sizes across figures.

Some of the major concerns:

  1. The main finding that CHIR99021 promotes lung epithelial cell proliferation and regeneration is not sufficiently novel given the existing literature showing impacts of this compound in lung progenitors.
  2. Key experiments demonstrating mechanistic links between CHIR99021 treatment and regeneration/repair outcomes are lacking. There is no strong evidence presented for causal relationships.
  3. The lack of anti-inflammatory effects of CHIR99021 conflicts with multiple previous studies and is not adequately explored. The basis for these differences requires deeper investigation.
  4. Functional readouts of lung injury and repair are missing - e.g lung mechanics and compliance measurements over time after treatment. Histological scoring alone is not enough.

Reviewer 3 Report

Comments and Suggestions for Authors

Dear authors,

As a reviewer, I appreciate the design and results of your paper.

1The article presents data on the effect of the non-selective inhibitor GSK3aß - CHIR99021 on the effects of sublethal doses of LPS on lung injury. The choice of drug and its dose are quite correct. There is a wealth of data in the scientific literature on the effect of different classes of GSK3 inhibitors on many human diseases, most notably diabetes, cancer, cardiac fibrosis and neurodegenerative diseases. In general, GSK3 inhibitors are currently considered to be anti-inflammatory factors. However, their effect may be contradictory, as the activity of GSK3 is regulated by many stress proteins and GSK3 in turn directly or indirectly regulates the function of up to 100 different proteins, including many transcription factors, cytoskeletal proteins, etc., involved in apoptosis, cell cycle control, the functions of some oncogenes, etc.

2. On the other hand, the effect of these drugs on acute critical conditions has not been studied in sufficient detail. This work is mainly represented by experimental studies. However, such studies do exist and they have not been fully analysed by the authors. Therefore, I suggest that the authors consider and analyse these works in more detail, including the possible use of GSK3 inhibitors in COVID-19.

3. Of course, the experimental model presented is not a full-fledged ARDS model in humans, but rather a model of a damaged lung. Therefore, I consider the authors' interpretations in this regard to be quite appropriate.

4. The authors used appropriate methods, including statistical methods. In addition to the main group, the study used a control group (without LPS) and a comparison group (LPS without CHIR). I think this is enough to get reliable results.

5. I think that the authors' interpretation of their own results in the discussion section is generally adequate. Perhaps the study could be briefly summarised in the conclusion section.

Currently, GSK3 inhibitors are already used to treat type 2 diabetes mellitus, neurodegenerative diseases and cancer. The role of GSK3 in ARDS is less clear. However, there are few studies using experimental models of damaged lungs. The authors have not fully reviewed these papers, including,PMC5308835.In addition, a more systematic analysis of the potential role of GSK-3 in the development of ARDS is presented in several narrative reviews, including PMC7446622.

Round 2

Reviewer 2 Report

Comments and Suggestions for Authors

Thank you for putting an effort in resolving comments.